*Perspective*

# Innate immunity in tumour immunoediting and immunosurveillance

Zhibin Zhang [1✉], Ying Zhang [2✉] & Judy Lieberman [3✉]

## Abstract

**The successes of cancer immunotherapy have inspired research aiming to increase the number of immune-responsive cancers. The first effective immunotherapeutic strategies—immune checkpoint blockade (ICB) and CAR T cells—were designed to overcome limitations in CD8[+] T cell recognition and killing of tumor cells. However, most solid tumors still do not respond to these measures and new treatment approaches are needed. Tumors evolve many strategies to avoid immune control. One way to identify immunotherapy strategies is to study what distinguishes immunotherapy-responsive and -unresponsive tumors. Another way is to identify the differences in tumors that emerge after carcinogen exposure in immunocompetent versus immunodeficient hosts. Still another way is to identify changes in gene expression in emerging tumors that enable them to escape immunosurveillance (known as tumor immunoediting). Evolving tumors suppress antigen processing and presentation to avoid triggering tumor-specific T cells but also repress key innate immune genes that transmit danger signals to immune cells. In this perspective, we discuss the roles of innate immunity in antitumor responses and consider how innate immunity could be harnessed to make tumors more immune-responsive.**

**Keywords** Innate Immunity; Immunoediting; Antitumor Immunity; Danger Signals; Immunotherapy
**Subject Categories** Cancer; Immunology

## Introduction

Immunotherapy has revolutionized cancer treatment, providing cures for some difficult-to-treat cancers (Baumeister et al, 2016; June et al, 2018; Sanmamed and Chen, 2018; Shah and Fry, 2019; Sharma and Allison, 2015). However, only a minority of solid tumors respond to immune checkpoint blockade (ICB; targeting immune checkpoint proteins, such as PD1, PD-L1 and CTLA4), and there are no chimeric antigen receptor (CAR) T cells approved for treating solid tumors (Baumeister et al, 2016; June et al, 2018; Morotti et al, 2021; Postow and Hellmann, 2018; Sanmamed and Chen, 2018; Shah and Fry, 2019; Sharma and Allison, 2015). Broadening the range of tumors that respond to immunotherapy is a crucial goal of cancer immunology research. To achieve this goal, cancer immunologists have tried to understand what distinguishes immunotherapy-sensitive tumors from immunotherapy-resistant tumors and have evaluated combinations of ICB antibodies or of ICB with other interventions clinically. However, recent efforts to increase ICB responsiveness have led to, at best, moderate improvements in response rates (Bonaventura et al, 2019; Breakstone, 2021; Jung et al, 2020).

Tumors use many strategies to avoid immune surveillance. Our understanding of the ways in which tumors sculpt the tumor microenvironment (TME) and evade immune recognition and of the many pathways and types of cells in the TME that suppress effective anti-tumor immunity has advanced considerably since the first cancer immunotherapy, which used a CTLA4-blocking antibody to suppress this inhibitory coreceptor on T cells, was clinically approved in 2011 (Hodi et al, 2010; Leach et al, 1996; Mellman et al, 2023). However, there is still much that we don't understand about how tumor immunity is regulated. Most of cancer immunology research and immunotherapy development that led to effective cancer immunotherapy has focused on harnessing or engineering cytotoxic T lymphocytes that detect antigens or cell surface-markers on tumor cells but not differentiated normal cells. However, both innate and adaptive immune cells also recognize signs of cellular or tissue stress or damage ("danger" signals). Tumor cells survive in immune competent hosts only if they avoid triggering the immune cells that can eliminate them. Danger signals that distinguish normal cells under homeostatic conditions from transformed tumor cells or cells that are infected with pathogens or damaged in other ways and need to be eliminated orchestrate a more potent and effective immune response that better activates killer lymphocyte functionality and memory. In this Perspective article, we suggest that one of the biggest obstacles to effective immune control of cancer is that tumor cells that are selected and edited to survive immune surveillance look a lot like normal untransformed cells to the immune system, making it difficult to distinguish the tumor from 'self'. Providing a 'danger' signal could increase immune recognition of tumors and thus broaden the range of tumors that respond to immunotherapy (Curtsinger and Mescher, 2010; Kroemer et al, 2024; Matzinger, 1994). However, the molecular features of many of the innate immune pathways that signal danger have only recently begun to be described and there is still a lot we don't understand about how

[1]Department of Immunology, University of Texas MD Anderson Cancer Center, Houston, TX, USA. [2]Key Laboratory of Cell Proliferation and Differentiation of the Ministry of Education, School of Life Sciences and Peking-Tsinghua Center for Life Sciences, Peking University, Beijing, China. [3]Program in Cellular and Molecular Medicine, Boston Children's Hospital, and Department of Pediatrics, Harvard Medical School, Boston, MA, USA. ✉E-mail: zzhang16@mdanderson.org; ying.zhang@pku.edu.cn; judy.lieberman@childrens.harvard.edu
https://doi.org/10.1038/s44318-025-00650-7 | Published online: 28 November 2025

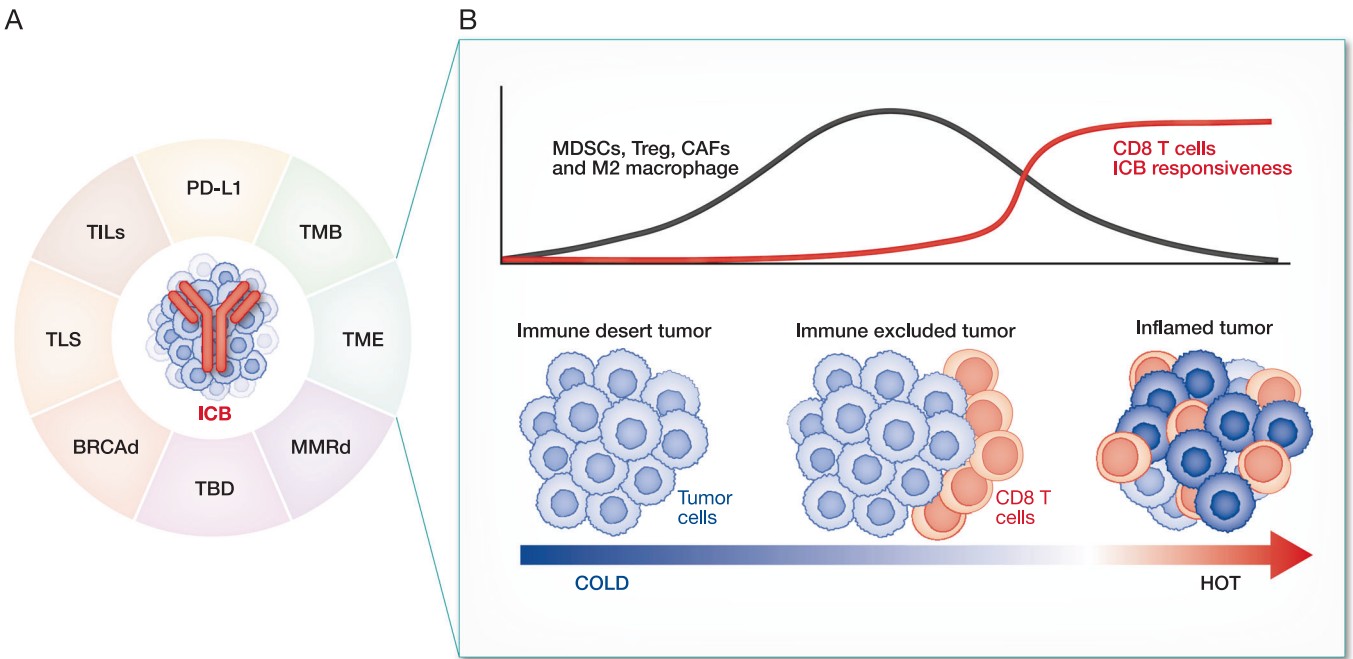

**Figure 1. Tumor properties that increase responsiveness to immunotherapy.**

(A) The determinants of tumor responsiveness to immunotherapy, particularly ICB, include: tumor mutational burden (TMB), mismatch repair deficiency (dMMR), BRCA deficiency (BRCAd), PD-L1 expression, tumor-infiltrating lymphocytes (TIL), tertiary lymphoid structures (TLS), inflamed or immunosuppressive tumor microenvironment (TME) and other factors yet to be determined (TBD). (B) TME states that are immunologically 'cold' and unresponsive to ICB ('immune desert' and 'immune excluded') and 'hot' (inflamed) tumors that may respond to ICB. Immune desert tumors lack tumor-infiltrating CD8[+] T cells. Immune-excluded tumors induce tumor-specific CD8 T cells but restrict their entry into the tumor. Instead, tumor-infiltrating immunosuppressive cells, including myeloid-derived suppressor cells (MDSCs), T regulatory (Treg) cells, cancer-associated fibroblasts (CAFs) and M2-like macrophages, accumulate in the TME. Inflamed tumors have infiltrating T cells in close contact with tumor cells.

they are triggered and regulated. The IFN response to tumors, which is mostly triggered by sensing features of viral and intracellular bacterial infection—especially mislocalized or modified nucleic acids in the cytosol—is well recognized as an effective danger signal in tumors, but further investigation is warranted to elucidate the role of other danger signals and innate immune and inflammatory pathways and to determine which are most effective at eliciting protective antitumour immunity. Activating these other innate immune and inflammatory pathways, such as NF-κB and pyroptosis, is a double-edged sword that may either increase tumourigenesis or bolster anti-tumor immunity (Grivennikov et al, 2010). For instance, during acute SARS-CoV-2 infection, a surge of IL-6 has been recently shown to awaken dormant tumor cells and drive cancer progression, a process facilitated by suppressive CD4[+] T cells within the tumor microenvironment (Chia et al, 2025). The potential for inflammation to promote cancer growth has

understandably made researchers cautious about harnessing these pathways for immunotherapy.

From an evolutionary perspective, however, the conflicting roles of inflammation on tumourigenesis and tumor immunity may not be surprising. Because cancers typically arise after reproductive age, immune control of cancer has not been under strong selective pressure. Cancer immunity instead relies on programs evolved for microbial defense and tissue repair. While the immune responses to pathogens and tissue damage can restrict tumors, the wound repair and other responses that promote tissue homeostasis often provide fertile ground for initiating tumors and promoting their growth. Importantly, the innate immune pathways triggered by cancer and the extent to which they can mount effective anti-tumor responses likely vary across tumor subtypes with distinct molecular features and at different stages of disease progression.

During early tumourigenesis, tumor cells that have mutated or changed their gene

expression to grow better, to survive in different tissues or hostile environments, or to avoid immune elimination preferentially survive (Khong and Restifo, 2002; Vesely et al, 2011). Here, we use the term 'tumor editing' to describe any alteration in the tumor genome or gene expression and reserve the term 'tumor immunoediting' to refer to changes in gene expression that allow the tumor to evade immune recognition and elimination. One way to identify strategies that could improve the response rate to immunotherapy is to understand better how tumors evade immune surveillance, as reversing tumor immunoediting could, in principle, restore immune surveillance. Previous studies that manipulated the expression of candidate immune genes suggested that tumor editing suppresses the expression of genes involved in T cell antigen expression, processing and presentation, which are crucial for adaptive immune responses against tumor cells (Khong and Restifo, 2002; Vesely et al, 2011). These early studies, combined with

---

**Box 1   Cytosolic DNA as a danger signal**

Mislocalized DNA in the cytosol is probably the most important danger signal in tumor cells. Cytosolic DNA could come from: (1) genomic DNA (gDNA) released from the nucleus as micronuclei, a byproduct of unrepaired DNA damage and chromosomal instability (CIN) (Krupina et al, 2021); (2) mitochondrial DNA released from mitochondria that are damaged during inflammatory cell death (de Torre-Minguela et al, 2021; Miao et al, 2023); or (3) reverse transcripts of endogenous retroelements, such as LINE-1 elements, that can be derepressed in poorly differentiated cancer cells (Kassiotis and Stoye, 2016; Lindholm et al, 2023).

Mislocalized DNA in the cytosol is a potent danger-associated molecular pattern that is recognized by multiple innate immune cytosolic sensors, including cGAS (Sun et al, 2013), IFI16 (Unterholzner et al, 2010), ZBP1 (Takaoka et al, 2007) and AIM2 (Fernandes-Alnemri et al, 2009; Hornung et al, 2009). In turn, these sensors can activate multiple innate immune pathways, resulting in the production of type I IFNs, NF-κB signaling, release of proinflammatory cytokines, pyroptosis and necroptosis. An example of tumors with cytosolic gDNA is the subsets of colorectal carcinoma (CRC) and endometrial, ovarian and gastric cancers with mismatch repair (MMR) deficiency—which can be caused by inherited mutations in MMR genes in the case of Lynch syndrome or sporadically owing to mutation or promoter methylation of the same genes. Unlike the corresponding MMR-proficient cancers, human MMR-deficient cancers respond to immune checkpoint blockade (ICB) (Cercek and Diaz, 2022; Germani and Moretto, 2021; Hewish et al, 2010; Le et al, 2015; Overman et al, 2018). MMR-deficient tumors accumulate cytosolic DNA, which can be detected as cytosolic puncta stained with DNA-binding dyes or by PCR analysis of cytosolic fractions.

The mechanisms mediating the inflammatory and immunogenic nature of MMR-deficient tumors are just beginning to be understood. MMR deficiency leads to instability of repeat sequences throughout the genome, which are prone to produce insertions or deletions during DNA replication, leading to microsatellite instability (MSI). MSI leads to the accumulation of unrepaired DNA damage, eventually driving chromosomal instability (CIN). Cells with CIN extrude DNA fragments in micronuclei whose membranes are unstable, causing micronuclei to release gDNA fragments into the cytosol. Thus, MMR deficiency not only increases tumor mutational burden (TMB) as a result of errors in DNA repair or translocations that produce tumor neoantigens, but also strongly activates inflammation in the tumor microenvironment, likely triggered by sensing cytosolic gDNA. MMR-deficient cancer cells have been shown to induce type I IFNs through activation of the cGAS–STING pathway by cytosolic DNA (Guan et al, 2021; Lu et al, 2021), but whether other DNA sensors or innate immune pathways are activated has not yet been studied. Human MMR-deficient, microsatellite-unstable CRC tumors are more inflamed than MMR-proficient, microsatellite-stable CRC tumors. MMR-deficient tumors markedly increase inflammatory gene expression in tumor-infiltrating macrophages and fibroblasts, are enriched for tumor-specific CD8[+] T cells, and have increased tumor cell expression of IFN-stimulated genes and chemokines (Pelka et al, 2021). Importantly, inflamed macrophages, fibroblasts and tumor cells colocalize with tumor-reactive CD8[+] T cells in foci that resemble secondary lymphoid tissue follicles. Given the strong correlation between DNA repair deficiency and enhanced responsiveness to ICB, understanding how genetic defects in DNA repair and accumulation of unrepaired DNA damage lead to tumor immunogenicity could guide new therapeutic strategies to make tumors more immunogenic. This will be facilitated by studies to determine the regulatory mechanisms that govern the activation of specific DNA-sensing innate immune pathways in various tumor contexts and to identify which DNA-sensing pathways are most effective in driving protective immunity.

---

studies that compared sarcomas that emerged in carcinogen-exposed immune competent vs deficient mice (Dunn et al, 2006; Schreiber and Podack, 2009), also suggested that genes that activate the type I interferon (IFNα and IFNβ) and type II interferon (IFNγ) pathways of the innate immune system are also suppressed during tumor editing (Dunn et al, 2006; Khong and Restifo, 2002; Vesely et al, 2011). Nevertheless, because tumors are usually only detected after they have undergone editing, until recently no unbiased studies have been carried out to systematically identify which genes are altered to allow a tumor cell to survive and grow in immunocompetent hosts. Genetically engineered mouse models (GEMM) of cancer and the development of unbiased "multiomics" techniques now provide an opportunity to study how emerging tumors evade immune control. Here, we discuss the implications of these models and other approaches that uncover how tumors survive immune elimination, focusing on the less well studied innate immune response to cancer and its potential relevance to designing the next generation of cancer immunotherapies.

## Tumor features that determine immunotherapy response

Studies in the past decade have used animal models and clinical samples, together with new unbiased 'omics' and gene editing screening techniques, to better characterize antitumour immunity and identify some of the features that distinguish cancers that respond or don't respond to ICB. Those tumors that are more responsive to ICB have higher tumor mutational burden (TMB), are deficient in mismatch repair (MMR) and other DNA repair pathways such as homologous recombination (HR), express the co-inhibitory checkpoint ligand PD-L1, or have more tumor-infiltrating lymphocytes (TILs), more tumor-specific T cells or more foci of tumor-specific T cells interacting with tumor cells and dendritic cells (DCs) and fewer immunosuppressive immune cells (Fig. 1A) (Barrett et al, 2015; Cercek and Diaz, 2022; Chen et al, 2024; Espinosa-Carrasco et al, 2024; Havel et al, 2019; Lyons et al, 2018; Pelekanou et al, 2018; Polk et al, 2018; Rousseau et al, 2021; Tomioka et al, 2018; Yarchoan et al, 2017). In a pan-cancer analysis of ICB-treated patients, ICB responsiveness was strongly linked to defects in multiple DNA repair pathways —MMR, HR, nucleotide excision repair, base excision repair and disruption of the proofreading function of DNA polymerases (Wang et al, 2018)—which result in increased TMB but also, importantly, an innate immune response to cytosolic DNA (Box 1). High TMB, PD-L1 expression, MMR deficiency or BRCA deficiency (which inhibits HR) are all used to select patients for ICB therapy (Ma et al, 2022; Samstein et al, 2021).

Unresponsive, immunologically 'cold', tumors include: (1) 'immune desert' tumors that lack TILs; and (2) 'T cell-excluded' tumors that stimulate CD8[+] cytotoxic T lymphocytes (CTLs) but prevent their entry into the tumor so that they align along the tumor margins (Hegde et al, 2016; Herbst et al, 2014) (Fig. 1B). 'Immune desert' tumors do not generate tumor-specific CD8[+] T cells, either because they lack effective T cell neoantigens or tumor-associated antigens or have edited them out by suppressing their expression or mutating the gene encoding the tumor antigen, MHC class I genes or other genes needed for antigen

processing and presentation. In 'immune desert' tumors and their draining lymph nodes, tumor-derived immunosuppressive factors (such as VEGF and TGFβ) may suppress the recruitment, maturation, and antigen processing and presentation capacities of DCs, which reduces the priming of tumor-reactive CD8[+] T cells (Joyce and Fearon, 2015; Wang et al, 2023). The mechanisms responsible for T cell exclusion from tumors are not fully understood but involve dysfunctional tumor vasculature (irregular structure, erratic or insufficient blood flow and high interstitial fluid pressure); Wnt–β-catenin and TGFβ signaling; other immunosuppressive molecules including IL-10, SOX10 and IDO; and tumor-infiltrating immunosuppressive cells, including regulatory T (Treg) cells (a source of TGFβ), myeloid suppressor cells, including M2-like macrophages and myeloid-derived suppressor cells (MDSCs), and cancer-associated fibroblasts (CAFs) (Joyce and Fearon, 2015; Wang et al, 2023). In some tumors, MDSCs and CAFs create a fibrotic TME, which impedes immune cell migration into the tumor. A better understanding of how T cell-excluded tumors are generated and the factors that mediate T cell exclusion will help to define strategies to make these tumors immunologically 'hot' and thus potentially more responsive to ICB.

Immunologically 'hot' tumors—only some of which respond to ICB—have tumor-specific, infiltrating T cells, often in close contact with tumor cells that can be organized into tertiary lymphoid structures (TLS) or immune foci with the capacity to activate T cell effector functions (Sautes-Fridman et al, 2019; Schumacher and Thommen, 2022). Although classical immunology describes that activation of naive antigen-specific T cells occurs in draining lymph nodes, recent studies strongly suggest that activation of tumor-specific T cells also occurs locally in the tumor. In fact, a recent paper suggests that the final activation of CD8[+] T cells to effector CTLs requires antigen- and costimulation-dependent activation within the tumor (Prokhnevska et al, 2023). ICB-responsive tumors have gene signatures that indicate more tumor-infiltrating CD8[+] T cells or natural killer (NK) cells and fewer immunosuppressive Treg cells and myeloid suppressor cells.

The mechanisms responsible for ICB resistance are complex and likely vary amongst tumor subtypes. Many immunosuppressive mechanisms remain to be uncovered. A recent intriguing example is a study that identified a new class of immunosuppressive CD4 + T cells in sarcoma-bearing mice that are distinct from Treg, express a distinct set of cell-surface and intracellular genes, and are differentially induced by vaccination with high dose versus low dose MHC-II tumor antigens (Sultan et al, 2024). Machine learning-based analysis of tumor features from patient biopsies and other clinical data will likely continue to improve our ability to predict which patients will respond to immunotherapy. However, a more mechanistic understanding of what underlies ICB resistance is needed to develop more effective therapies for a larger proportion of tumors.

In immunologically 'hot' tumors, unlike 'cold' tumors, the tumor cells, immune cells and stromal CAFs are often 'inflamed', expressing genes indicative of sensing some sort of danger signal within the tumor. In fact, the only colorectal carcinomas (CRC) that respond well to ICB are the small subset of cancer that are MMR-deficient, which is linked to microsatellite instability (MSI) and chromosomal instability (CIN) (Hewish et al, 2010; Le et al, 2015). Of note, the TME of MMR-deficient, microsatellite-unstable, ICB-sensitive human CRC—unlike MMR-proficient, microsatellite-stable, ICB-insensitive human CRC—contains foci of tumor cells interacting with TILs and stromal cells, all of which have inflammatory gene signatures (Pelka et al, 2021). CIN developing from MSI in MMR-deficient CRC leads to formation of micronuclei, which can release genomic DNA fragments into the cytoplasm; this triggers a type I IFN response through the cGAS–STING pathway (Harding et al, 2017; Mackenzie et al, 2017). It might also activate other pattern recognition receptors that recognize mislocalized cytosolic DNA (such as AIM2, IFI16 and ZBP1) (Box 1) and trigger inflammatory cell death (pyroptosis and necroptosis) (Vanpouille-Box et al, 2018), but these pathways have not yet been studied in the context of tumoral DNA damage repair defects (Box 1).

## The role of danger signals in tumor immunogenicity

The prevailing view about why the immune system fails to control cancer is defective adaptive T cell-mediated immunity. Indeed, many studies, including adoptive transfer and antibody depletion experiments, have clearly shown that cytotoxic CD8[+] T cells are the key effector cells responsible for immune control of cancer (Dunn et al, 2002; Dunn et al, 2004). Their critical role is underlined by the successes of immunotherapy—CD8[+] T cells are the primary target for ICB therapy and mimicking them is the goal of CAR-T cell therapy. For some tumors, tumor antigen-specific CD8[+] T cells are not stimulated or are repelled from entering the tumor, but for others, tumor-specific T cells are present in the tumor but are dysfunctional. T cell dysfunction, a hallmark of chronic infection and cancer where antigens persist without adequate costimulation and CD4 T cell help, arises through multiple mechanisms. In their initial activation, CD8[+] T cells transiently induce checkpoint inhibitory receptor genes to restrain excessive cytotoxicity and cytokine release, but with chronic stimulation these inhibitory pathways are sustained and contribute to CD8[+] T cell dysfunction. Besides checkpoint signaling, CD8[+] TIL also have impaired effector functions, including reduced expression of cytotoxic effector molecules (perforin and granzymes), defective TCR signaling, and diminished production of IL-2 and other cytokines (Trimble and Lieberman, 1998). These intrinsic defects are reinforced by extrinsic pressures within the TME: nutrient deprivation and metabolic competition driven by the Warburg effect, accumulation of lactic acid and hypoxia, insufficient CD4[+] T cell help, and exposure to immunosuppressive cytokines such as TGF-β (Donkor et al, 2011) and IL-10. Moreover, the TME is enriched with immunosuppressive cell populations—including myeloid-derived suppressor cells, neutrophils, dysfunctional dendritic cells, Treg, and cancer-associated fibroblasts (CAFs)—that further dampen CD8[+] T cell activity. Together, these factors establish a profoundly suppressive environment that drives and maintains T cell dysfunction. A discussion of what is known about the factors that contribute to T cell dysfunction, and how tumor cells sculpt the TME to recruit and redirect hematopoietic cell differentiation and functionality to orchestrate immunosuppression to survive immune elimination, is beyond the scope of this review (see (Thommen and Schumacher, 2018)).

The crucial final effector cells responsible for effective immune surveillance are tumor-specific cytotoxic lymphocytes, mostly CD8[+] CTLs but also other types of cytotoxic lymphocytes (NK cells, innate lymphoid cells (ILCs), γδ T cells,

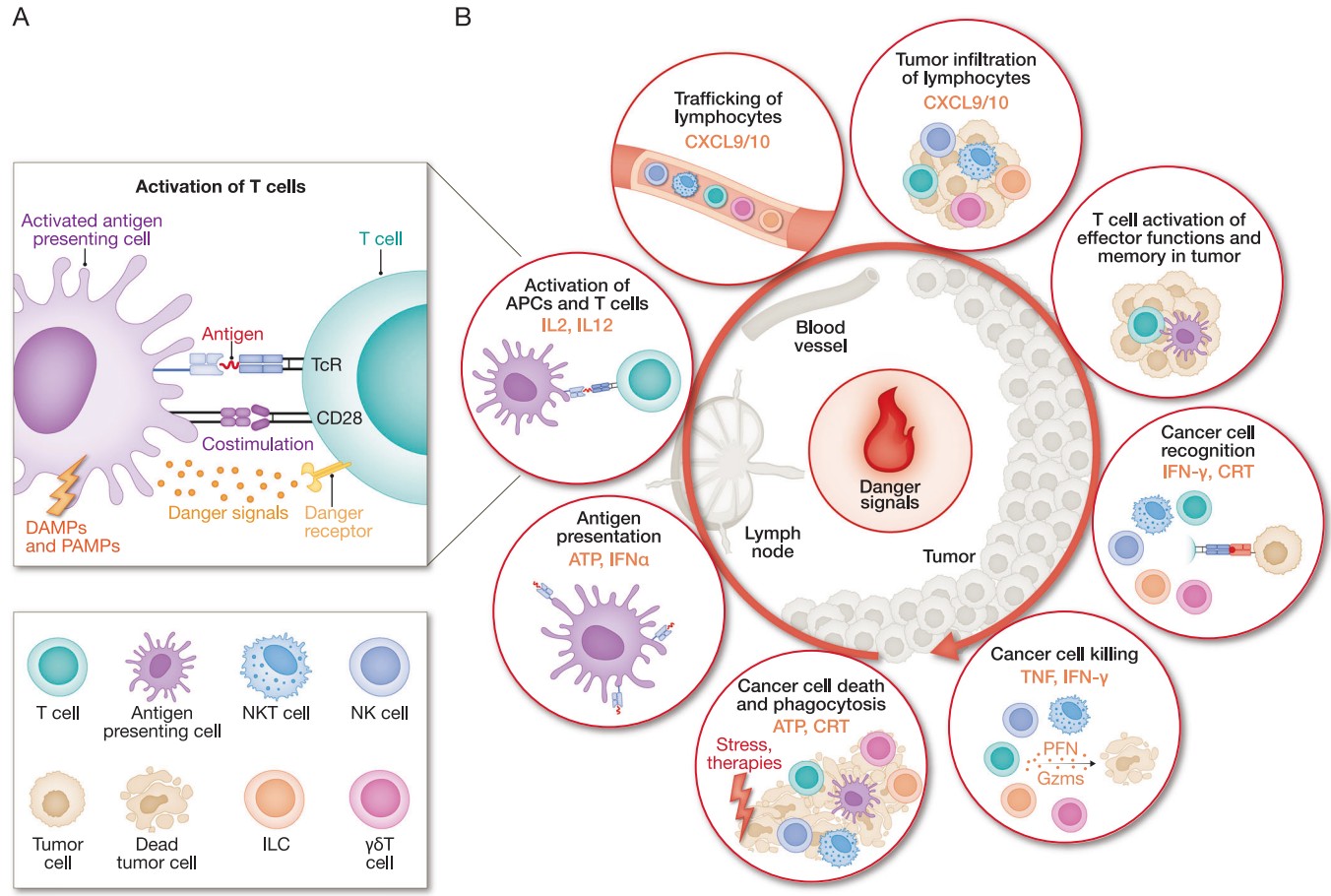

**Figure 2. Danger signals are required to fully activate protective T cell immunity.**

(A) T cell activation by antigen-presenting cells (APCs) requires three signals: (1) antigen recognition by the T cell receptor (TCR); (2) a costimulatory signal to activate CD28; and (3) cytokines or other danger signals. Danger- or pathogen-associated molecular patterns (DAMPs or PAMPs) prime and activate APCs. (B) Danger signals, including cytokines, chemokine and alarmins (DAMPs and PAMPs), promote the cancer–immunity cycle (modified from Mellman et al (2023)) and are needed to establish T cell functionality and memory. Representative danger signals that promote individual steps of the cycle are highlighted in orange. In addition to CD8 T cells, cancer cells can be recognized and killed by innate-like lymphocytes such as γδ T cells, NK cells, ILCs, and NKT cells. Cancer cells also undergo cell death in response to cancer therapies (e.g., radiotherapy, targeted therapy, chemotherapy and immunotherapy) and stress within tumors (e.g., ER stress, genotoxic stress, hypoxia stress, oxidative stress, chromosomal instability and unrepaired DNA damage).

NKT cells, MAIT cells and CD4$^+$ T cells). The relative contributions of different types of cytotoxic lymphocytes to tumor elimination may vary between tumors. The full activation of T cells to kill tumor cells requires 3 signals: T cell receptor (TCR) binding to an antigenic peptide presented by class I MHC to activate TCR signaling ('signal 1'); costimulation (activated by binding of CD28 on the T cell to its activating ligands CD80 and CD86 ('signal 2')) to trigger clonal expansion; and a danger signal ('signal 3') (Fig. 2A). The effects of both CAR T cells and ICB depend on the activation of CD8$^+$ CTLs to kill the tumor, either by bypassing the TCR signal ('signal 1') and replacing it with transferred T cells that express a CAR that binds to a tumor-specific cell surface ligand and is

fused to activating intracellular signaling domains, or by counteracting inhibitory signals that block T cell costimulation ('signal 2'). However, there are other ways, besides interfering with signal 1 and signal 2, that tumors use to evade immune surveillance. A danger signal is needed to activate effector cell functions and generate antitumour memory lymphocytes (Curtsinger and Mescher, 2010; Kroemer et al, 2024). This signal is often designated in a general manner as a cytokine input such as type I IFNs or IL-12, but its precise nature and its context-dependent requirements remain incompletely defined. Danger signals may arise directly from tumor cells, from antigen-presenting cells that engulf dying tumor cells, or from soluble mediators released by other cells in the TME.

How tumor cells die strongly affects danger signaling. Apoptosis is typically non-inflammatory and fails to provide a danger signal. However, some cytotoxic therapies that cause apoptosis also trigger ICD. Chemotherapeutics, such as mitoxantrone, oxaliplatin and doxorubicin, as well as γ-irradiation, induce ER stress during apoptosis, which activates macrophage phagocytosis and NK recognition and killing of tumor cells (Obeid et al, 2007; Sen Santara et al, 2023). By contrast, programmed necrotic cell death, such as pyroptosis, necroptosis and possibly ferroptosis, trigger both inflammation and immunogenic cell death (ICD). They rupture the plasma membrane and trigger the release of cytokines, chemokines and alarmins such as ATP and HMGB1. These signals recruit and

## Box 2 Immunogenic cell death

Some dying cancer cells elicit an immune response in vivo that can protect mice from subsequent implantation with untreated viable cancer cells. The type of tumor cell death is defined as immunogenic cell death (ICD) (Galluzzi et al, 2017a; Galluzzi et al, 2020). The gold-standard assay that defines what types of treatment cause ICD is protection against secondary tumor challenge after vaccination with dying tumor cells. ICD was originally identified in tumors treated with radiotherapy and a subset of cytotoxic chemotherapy drugs, which usually induce apoptosis. It was originally proposed as a mechanism by which the immune system eliminates the small population of tumor cells that survive cytotoxic drugs and thereby blocks tumor recurrence (Casares et al, 2005). A potent 'danger' signal that activates innate immunity is generated by cancer cells dying by ICD. ICD is linked to treatments that cause ER stress in the tumor, which results in cell-surface externalization of the ER-resident chaperone, calreticulin (Obeid et al, 2007). Externalized calreticulin is recognized as a danger signal by both the low-density lipoprotein-related protein (LRP) receptor on macrophages, which triggers phagocytosis (Gardai et al, 2005; Obeid et al, 2007), and the natural killer (NK) cell-activating receptor NKp46, which triggers NK cell-mediated killing and cytokine secretion (Sen Santara et al, 2023). Inflammatory forms of programmed cell death (pyroptosis, necroptosis and probably ferroptosis) also induce ICD (Aaes et al, 2016; Efimova et al, 2020; Zhang et al, 2020), but it remains to be shown whether these inflammatory cell death programs also cause calreticulin externalization. Cells undergoing necrotic cell death release danger signals, known as alarmins, which include inflammatory cytokines, chemokines and other danger-associated molecular patterns, such as ATP and HMGB1. These signals recruit and activate immune cells that both eliminate the tumor and establish immune memory. ICD is not only triggered by exogenous chemotherapy agents and radiation but also can occur spontaneously in some tumors. This can occur in areas of hypoxia or nutrient deprivation, in senescent tumor cells and during cytotoxic lymphocyte attack that results in the formation of pyroptosis-inducing, gasdermin pores in the tumor cell membrane. In addition, some tumors, such as lymphomas that undergo ER stress because of high levels of protein synthesis, are also prone to spontaneous ICD.

Programmed necrotic forms of cell death—pyroptosis and necroptosis, and possibly ferroptosis (although this is controversial (Efimova et al, 2020; Wiernicki et al, 2022))—cause immunogenic cell death (ICD) (Aaes et al, 2016; Efimova et al, 2020; Zhang et al, 2020). Pyroptosis is defined as cell death mediated by pore-forming gasdermins (GSDMs), of which there are 5 in humans (GSDMA to GSDME) (Ding et al, 2016; Kayagaki et al, 2015; Liu et al, 2016; Shi et al, 2015). The most studied of these, GSDMD, is activated by inflammasome sensors of microbial PAMPs or sterile DAMPs in the cytosol (Broz and Dixit, 2016; Zheng et al, 2020). The canonical inflammasomes (formed by NOD-like receptors, AIM2-like receptors and pyrin) recruit and activate caspase-1, which cleaves and activates GSDMD as well as the pro-forms of the IL-1-family cytokines to their active state (Martinon et al, 2002). The IL-1-family cytokines, which cause fever and are arguably the most inflammatory cytokines in the body, do not have a secretion signal and their release from activated cells generally requires GSDM pores (Heilig et al, 2018; Xia et al, 2021). The canonical inflammasomes and their downstream mediators are expressed at sites such as the mucosal epithelia and skin that are most likely to be exposed to infection, and in myeloid sentinel cells (macrophages, monocytes and dendritic cells) in the blood and most tissues, and their expression is primed when cell-membrane sensors such as the Toll-like receptors sense microorganisms.

The non-canonical inflammasome pathway is mediated by human caspase-4 and caspase-5 and mouse caspase-11 (Hagar et al, 2013; Kayagaki et al, 2011; Kayagaki et al, 2013). These caspases sense lipopolysaccharides (LPS) of Gram-negative bacteria and possibly other microbial lipids as well as oxidized host lipids (Shi et al, 2014; Zanoni et al, 2016). When they bind to LPS, the pro-forms of caspase-4, caspase-5 and caspase-11 are autoproteolyzed and activated to cleave GSDMD to trigger pyroptosis (Shi et al, 2014). Human caspase-4 and caspase-5, but not mouse caspase-11, also proteolyze and activate pro-IL-18, and all three caspases secondarily activate the canonical NLRP3 inflammasome and hence caspase-1 to cleave and release IL-1β and IL-18 (Baker et al, 2015; Devant et al, 2023; Ruhl and Broz, 2015; Schmid-Burgk et al, 2015; Shi et al, 2023). Caspase-4 is constitutively expressed in human endothelial cells together with pro-IL-18 and GSDMD and its activation can lead to vascular leak (Wei et al, 2024).

Pyroptosis can also be triggered by other proteases besides the inflammatory caspases and by other GSDMs (Liu and Lieberman, 2024; Liu et al, 2021). The apoptotic caspases can activate pyroptosis in GSDM-expressing cells—for example, caspase-3 can cleave GSDME (Rogers et al, 2017; Wang et al, 2017), and caspase-8 can process both GSDMC (Hou et al, 2020; Zhang et al, 2021a) and GSDMD (Orning et al, 2018; Sarhan et al, 2018) in distinct contexts to initiate pyroptosis. Also, the granzyme A and B serine proteases that mediate killing by cytotoxic T lymphocytes can cleave and activate GSDMB (Zhou et al, 2020) and GSDME (Zhang et al, 2020), respectively, to trigger pyroptosis. Expression of these GSDMs is frequently repressed in tumors as a mechanism of immune evasion (de Beeck et al, 2012; Kong et al, 2023; Liu et al, 2021; Zhang et al, 2020).

Some myeloid cells that are triggered to undergo pyroptosis can repair their membrane damage and survive and release cytokines at increased levels. These cells are termed 'hyperactivated' (Zanoni et al, 2016; Zanoni et al, 2017). Vaccination with hyperactivated dendritic cells (DCs) more effectively induces protective antitumour immunity in mouse models of immunologically cold tumors than vaccination with apoptotic or pyroptotic dying tumor cells (Zhivaki et al, 2020), which suggests the importance of DC-secreted danger signals in induction of antitumour immunity.

Necroptosis is mediated by the RIPK1–RIPK3–MLKL signaling pathway downstream of multiple immune receptors (Galluzzi et al, 2017b; Sun and Wang, 2014), including TNFR1 for tumor necrosis factor, the LPS receptor TLR4, and ZBP1 detecting double-stranded DNA. MLKL translocation to the plasma membrane is required for membrane rupture during necroptosis (Cai et al, 2014; Dondelinger et al, 2014; Wang et al, 2014), but the underlying mechanism remains less clear, although it is thought to be mediated by ion channels formed by phosphorylated MLKL. Necroptosis has been implicated in both tumourigenesis and tumor protection (Najafov et al, 2017). Necroptosis is less inflammatory than pyroptosis because it does not directly activate processing of pro-IL-1 family cytokines.

---

activate immune cells, drive the formation of tumor-reactive memory T cells and promote long-term tumor control (Aaes et al, 2016; Efimova et al, 2020; Zhang et al, 2020) (Fig. 2B, Box 2). Pyroptosis is especially inflammatory. Cytotoxic T lymphocytes (CTLs), which normally cause noninflammatory programmed cell death, can trigger pyroptosis when tumor cells express gasdermins (GSDMs) (Zhang et al, 2020; Zhou et al, 2020). Granzymes A and B cleave GSDMs directly: granzyme B activates GSDME both directly and indirectly through caspase-3, while granzyme A activates GSDMB. Activated GSDMs form pores in mitochondrial and plasma membranes, leading to cell rupture and the release of inflammatory mediators (Ding et al, 2016; Liu et al, 2016; Miao et al, 2023). Many tumor cells evade this pathway by epigenetically silencing GSDME (de Beeck et al, 2012).

Cytosolic DNA is an especially important danger signal for tumors. It is sensed by cytosolic sensors that can activate both type I IFN pathways and inflammatory programmed cell death (Box 1). Inflammatory death is likely to generate a stronger alarm than IFN signaling, even though the type I IFNs activate NF-κB, a master inflammatory transcription factor. Pyroptosis is expected to be more inflammatory than necroptosis, as inflammatory caspases not only activate

gasdermins but also process IL-1 family cytokines, which then exit through gasdermin pores. Critical questions remain. Which inflammatory pathways are activated in tumor cells with DNA repair defects or other inducers of cytosolic DNA? What determines the choice of pathway? How do these pathways shape antitumour immunity and affect tumourigenesis? Addressing these questions may reveal strategies to exploit these signals to enhance immunotherapeutic efficacy.

Although danger signals recruit and increase the functionality and memory of protective immune cells (in particular, CD8 + T cells) in the TME, they can also increase cancer cell-intrinsic malignant properties and promote an immunosuppressive TME, especially if they are chronically active. This seemingly paradoxical feature of chronic innate immune stimulation has been described for Type I and Type II IFN signaling. It may have arisen by evolutionary selection in response to persistent pathogens to limit immune activation to avoid immune pathology. Tumor-generated "danger" signals elicited by cytosolic DNA can promote the recruitment, development and survival of suppressor cells, especially MDSCs, in the context of chronic inflammation (Zhao et al, 2021). Chronic activation of the cGAS-STING pathway in some CIN tumor models leads to tumor suppression and increased metastasis in some models (Bakhoum, 2022; Li et al, 2023b). With chronic cGAS-STING activation, the induction of Type I IFNs or ISGs stops but other induced genes continue to be expressed, which create a more immunosuppressive TME. The balance between protective and suppressive effects of danger signals likely varies widely across cancer subtypes and disease stages, making it highly complex. Defining the contexts in which danger signals promote protective immunity without increasing immune suppression is an important area for future investigation.

## Potential roles of innate and innate-like lymphocytes in tumor elimination

In this section, we briefly discuss how innate immune lymphocytes or conventional T cell activation by NK receptors can also potentially contribute to tumor elimination. Innate and innate-like lymphocytes are not abundant in most tumors, suggesting that in most settings they do not play a major role in immune protection, but that does not exclude the possibility that some of them (especially NK) might help control some emerging tumors before adaptive immunity has a chance to develop or might orchestrate the type of adaptive immune response that develops or that they might be harnessed therapeutically. One example are the preclinical and clinical efforts to design effective CAR-NK cells, although these have mostly focused on hematological cancers, where CAR-T cells have been effective and are FDA-approved (Dagher and Posey, 2023). The low numbers of NK and innate-like lymphocytes in the TME also means that innate and innate-like cytotoxic lymphocytes are rarely analyzed in "omics" studies.

Tumor cells are often subject to various forms of cellular stress (such as hypoxia, genotoxic stress, CIN, metabolic and ER stress). The innate immune system is uniquely equipped to recognize and eliminate stressed cells independently of tumor-specific antigens. The activating receptors of innate and innate-like lymphocytes (NK cells, γδ T cells, NKT cells, MAIT cells and ILCs) often recognize signs of cellular stress and thus provide a mechanism to eliminate tumor cells that may not express tumor-specific antigens and to control the growth of tumors before antigen-specific effector T cells have been generated (Benci et al, 2019; Chiossone et al, 2018; Ruf et al, 2023). Antibody depletion and knockout studies have shown that lymphocytes that lack TcRαβ or mice that lack class I MHC or tumors that lack antigen processing and presentation machinery still show anti-tumor immunity (DuPage et al, 2012; Girardi et al, 2001; Grusby et al, 1993). In mice and in human cells cultured in vitro, CD8+ TIL can kill MHC-I–deficient tumor cells through the NKG2D–NKG2DL axis, enabling antigen-independent but priming-dependent cytotoxicity (Lerner et al, 2023). Recent studies in autoimmune mice also show that a subset of resident memory T cells ($T_{RM}$) can also express some NK activating receptors and NK costimulatory receptors, suggesting that they might be able to recognize and eliminate stressed cells independently of the TCR (Koh et al, 2023). Studies from the Ming Li laboratory showed that lymphocytes with NK-like properties are also present in a mouse breast cancer model (Chou et al, 2022; Dadi et al, 2016). The mammary glands of 8-week-old female GEMM mice expressing an MMTV-driven simian virus 40 T antigen (PyMT mice), who had very early-stage tumors that were not yet palpable, identified expansions of infiltrating granzyme B+ lymphocytes that also expressed resident memory cell adhesion molecules (i.e., CD49a and CD103) and repressed chemokine receptors and other molecules associated with circulating lymphocytes. Although these TILs belong to 3 distinct lineages, TcRαβ+, TcRγδ+ and TcR- (ILC1-like), they have very similar gene expression patterns, which suggested that they are proliferating and cytotoxic. Indeed, these TILs are PD-1-, proliferate and kill PyMT-derived target cells and their expansion depends on IL-15. Importantly, they also kill target cells that lack antigen processing, suggesting that they are capable of TCR-independent killing. Indeed, the TCRαβ+ TIL with NK-like properties are polyclonal and do not show skewed TCR repertoires, as would be expected if they were recognizing conventional antigens through their TCRs. These cells express many of the same genes as NK cells— perforin as well as NK activating and inhibitory receptors, signaling molecules and cytotoxic cell transcription factors, characteristic of NK cells. These TIL with NK properties might play an important role in eliminating tumor cells, both in emerging tumors and established tumors, especially those that have repressed antigen processing and presentation genes during immunoediting. It is worth investigating whether innate-like T cells are generated in other types of cancer and in humans, the mechanisms responsible for their generation and anti-tumor functions, whether they become depleted or dysfunctional during tumor progression, and developing therapeutic strategies to enhance or mimic this type of innate immune response.

NK cells and innate-like killer lymphocytes in the TME can also help to control tumor growth after tumor-specific T cell exhaustion. However, like T cells, NK cells can become exhausted in established tumors (Bi and Tian, 2017; Ruf et al, 2023). Myeloid cells phagocytose stressed and dying tumor cells to both eliminate tumor cells and present their antigens to T cells to stimulate adaptive immunity. Enhancing these effector functions of innate immune cells may be an effective approach to broaden tumor immunity. However, a molecular understanding of the innate immune stress receptors, what they recognize as signs of stress and the signaling pathways induced by different forms of cellular stress in

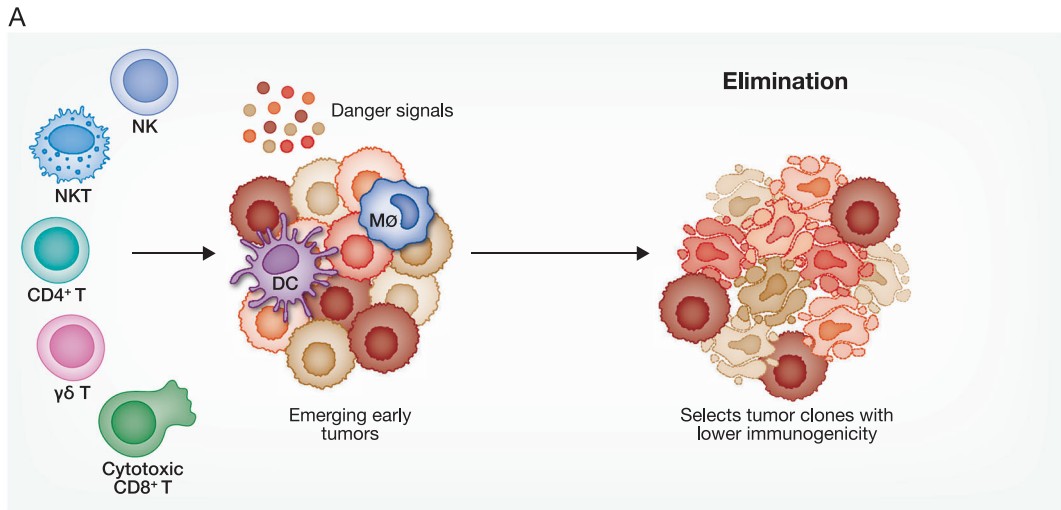

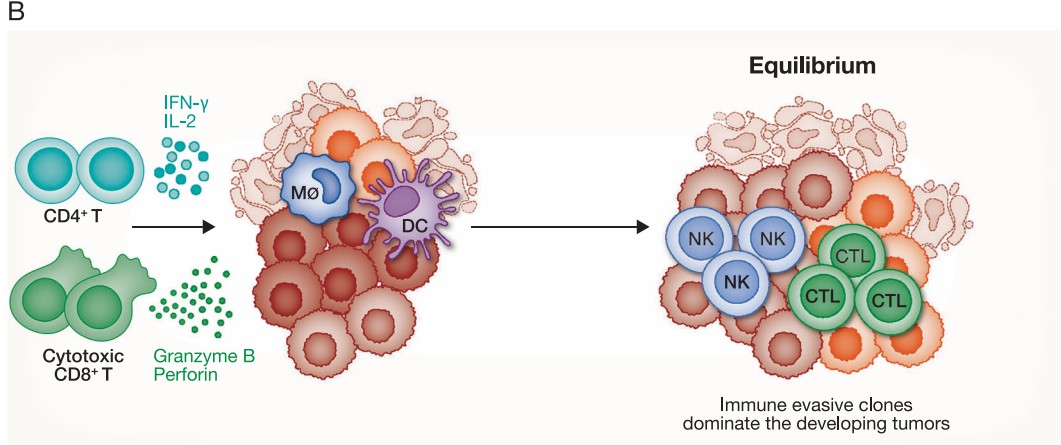

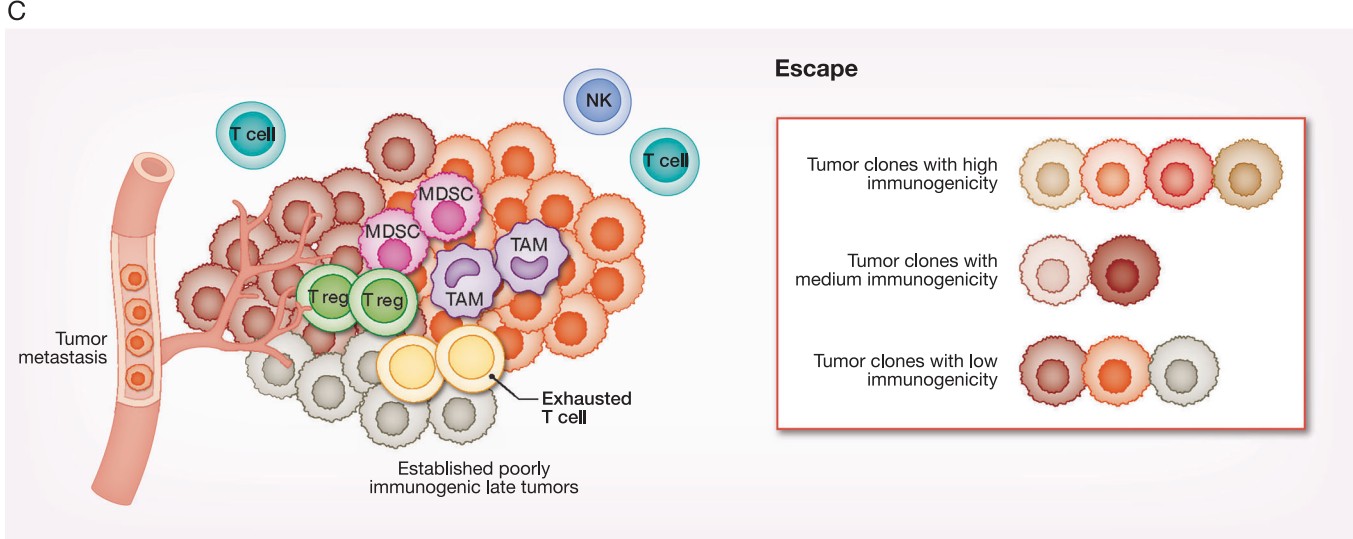

tumor cells is still limited. For example, stress-induced cell surface ligands on uninfected cells have been identified for only two of the human NK cell activating receptors (which are needed to activate NK cell-mediated killing and cytokine secretion): (1) MICA, MICB and ULBP1 to ULBP6 ligands for NKG2D, which are often expressed by cancer cells and induced by a not-well-defined variety of stresses, including infection, heat shock, genotoxic stress and IFNγ (Groh et al, 2002; Salih et al, 2002); and (2) the ER-resident chaperone calreticulin as a ligand for NKp46, which is expressed on all NK cells and some ILCs (Sen Santara et al, 2023). Cell surface

**Figure 3. Tumor immunoediting.**

(A) During the elimination phase of tumor development, early tumors contain tumor cells of varying immunogenicity. A fraction of tumor cells succumb to metabolic stress or attacks from innate immune cells, releasing danger signals that recruit both innate and adaptive immune cells into the TME. For some tumors, this initial response orchestrates an immune response that eliminates highly immunogenic tumor cells. However, less immunogenic clones can evade immune surveillance, survive, and replicate. (B) Surviving, poorly immunogenic tumor cells persist in the equilibrium phase, where tumor growth is contained by immune cells. Innate and adaptive lymphocytes suppress tumor growth by releasing the contents of cytotoxic granules that contain granzymes and perforin and by secreting cytokines (such as IFNγ and TNF). Tumor-infiltrating myeloid cells phagocytose stressed tumor cells and activate antitumour innate and adaptive lymphocytes. In this phase, some tumor cells may enter a quiescent state that is resistant to therapy and immune attack. Others proliferate and are selected for mutations that enable immune escape. In this phase, the immune response continuously eliminates emerging immunogenic clones, maintaining tumor control without eradicating the tumor. (C) Poorly immunogenic tumor clones eventually evade immune control and grow unchecked. In this escape phase, tumor cells undergo immunoediting that develops additional strategies to avoid antitumour immune responses. These strategies include downregulating tumor antigen expression and antigen presentation, secretion of immunosuppressive factors (such as TGF-β, IL-10), recruitment of immunosuppressive cells, such as Treg cells and MDSCs, and suppression of tumor danger signals. In addition, tumor cells exploit immune checkpoints, by expressing immune checkpoint receptor ligands, such as PD-L1.

calreticulin is also an important trigger of myeloid cell phagocytosis of ER-stressed cells (Gardai et al, 2005). However, how soluble calreticulin, which is retained in the ER under non-stressed conditions, traffics to the cell membrane during ER stress is not well understood. Only a small proportion of cellular MICA and MICB is found on the cell surface where it would be needed to activate NKG2D; its cell surface abundance is reduced by proteolytic cleavage and shedding (Groh et al, 2002; Salih et al, 2002) but the pathways that control its trafficking and exposure on the cell membrane are not fully understood. A better understanding of how cellular stress is sensed by NK cells and other classes of innate-like cytotoxic lymphocytes and phagocytic myeloid cells could help to harness these alternate pathways for cancer immunotherapy and to improve the adjuvants used in cancer vaccines.

## The role of tumor editing in immune escape

The concept of immunoediting of emerging tumors was presciently elaborated more than twenty years ago by Robert Schreiber and colleagues (Dunn et al, 2002; Dunn et al, 2004), at a time when few cancer researchers thought that immunosurveillance of tumors existed, except possibly for virally mediated cancers. They posited that emerging tumors are initially recognized and controlled by immune surveillance (Fig. 3A), as had been proposed by Frank Macfarlane Burnet and Lewis Thomas (Burnet, 1957; Thomas, 1959), but that tumors change their gene expression to evade immune recognition and survive (Fig. 3B,C). Their model of cancer immunity proposed that innate and innate-like lymphocytes, including NK cells, γδ T cells and NKT cells, initially recognize newly

transformed cells. These lymphocytes kill some tumor cells and produce cytokines, including IFNγ, that activate macrophages and other antigen-presenting cells, and chemokines that recruit immune cells to the tumor. Activated macrophages and DCs phagocytose dying tumor cells and present tumor neoantigens and tumor-associated antigens; they traffic to lymph nodes, where they activate tumor-specific T cells that home to the tumor and kill the tumor cells. Immunoediting enables tumor cells to evade T cell recognition and elimination. In keeping with this model, early studies showed that tumor editing represses antigen processing and presentation pathways and the expression of class I MHC to interfere with conventional TCR recognition, thus eliminating 'signal 1' for CD8$^+$ T cell activation and its downstream antitumour effects (Khong and Restifo, 2002; Vesely et al, 2011). The underlying mechanism(s) responsible for immunoediting were left unspecified in the original papers but could involve genetic mutations and copy number variations (CNV), as well as epigenetic changes to alter gene expression.

Immune responses to cancer were initially proposed to consist of three phases: elimination (immune surveillance, which purges tumor cells that are killed or phagocytosed) (Fig. 3A), equilibrium (immune restraint of tumors that escape surveillance) (Fig. 3B) and escape (tumor outgrowth and clinical disease) (Dunn et al, 2002; Dunn et al, 2004) (Fig. 3C). Cancer immunoediting could, in principle, occur at all stages of tumourigenesis, but tumor cell gene expression has mostly been studied in established tumors without comparing them to early tumors, which are difficult to capture in adequate numbers. In fact, there are no good estimates of what proportion of emerging tumors in humans are effectively

eliminated or initially controlled by immune surveillance. However, a comparison between early and established tumors can distinguish changes in tumor gene expression that are required for cellular transformation from changes that are needed to evade immune control. One way to bypass the difficulty of capturing enough cells in nascent tumors to identify immune-edited genes is to compare gene expression in spontaneously arising or carcinogen-induced tumors in immunocompetent mice versus profoundly immunodeficient mice (for example, NOD–SCID IL2Rγ-null (NSG) mice or perforin-deficient mice) where adaptive immune selection pressure is minimal and innate immune selective pressure is partially crippled.

The importance of innate immunity in driving cancer immunoediting, even without adaptive immunity, was established in a study more than ten years ago by comparing spontaneously arising sarcomas induced by methylcholanthrene in wild-type mice, adaptive immune-deficient *Rag2*$^{-/-}$ mice and *Rag2*$^{-/-}γc^{-/-}$ mice, which lack all lymphocytes (O'Sullivan et al, 2012). Tumors that arose in *Rag2*$^{-/-}γc^{-/-}$ mice in the absence of lymphocyte selection pressures were more immunogenic than tumors that arose in *Rag2*$^{-/-}$ mice lacking just adaptive immunity or in wild-type mice proficient in both innate and adaptive immunity, and these immunogenic tumors were better controlled when implanted in wild-type mice. This study also suggested that innate immune protection against tumor cells in the absence of adaptive immunity was mediated by NK cells and their secretion of IFNγ.

The immune pathways that regulate clinically important transitions in tumourigenicity and malignancy, including local invasion, colonization of distal tissues and

activation of dormant tumor cells to proliferate and form macroscopic metastases, are still mostly not well understood but understanding them better could have important implications for deciding which patients to treat and identifying new immune therapeutic targets. Tumor cells that have seeded distant metastatic sites and are dormant (in equilibrium) undoubtedly are under immune control until they escape from the immune response to generate macroscopic metastases (Giancotti, 2013; Kumar et al, 2024; Malladi et al, 2016; Massague and Obenauf, 2016). Thus, tumor immunoediting also likely controls the clinically important transition from dormancy to proliferation and macrometastasis, but the steps involved in orchestrating this transition are not well described. Multiomics technologies, especially those that provide temporal and spatial snapshots of both emerging and metastatic tumors can identify candidate genes and pathways to better understand the role of innate and adaptive immunity in controlling tissue colonization and dormancy at metastatic sites and understand how the tumor manipulates the metastatic niche and TME. Human cancer biopsies sometimes contain coexisting areas of locally contained and invasive tumors (e.g., breast cancer ductal in situ carcinoma and invasive lesions) (Casasent et al, 2018; Wang et al, 2024); spatial genomics, transcriptomics and multiplexed proteomics imaging methods are powerful methods to begin to identify what differentiates invasive lesions (Casasent et al, 2018; Sinha et al, 2021; Wang et al, 2024).

## Insights about tumor immunoediting from genetically modified mouse tumor models

Genetic manipulation of oncogenes and tumor suppressor genes in mice is a powerful tool to systematically identify the changes in gene expression occurring in emerging tumors that enable them to escape immune surveillance and to define how editing affects the immune response to the tumor. In principle, genetically manipulated mice could be used to capture not only transformed cancer cells but also premalignant cells expressing oncogenes prior to transformation (Kang et al, 2011). GEMM tumors are good models for studying immunoediting because they arise in and receive signals from the TME, and they

more closely resemble human cancers than cancer cell lines (which have already undergone immunoediting to escape immune surveillance). Because GEMM can be synchronized to develop aggressive cancers rapidly using inducible oncogenes that drive human cancer and the transformed cells can be readily identified by co-inducing a fluorescent marker gene, they are experimentally practical for capturing early events in tumorigenesis as well as the gene editing and selection that occurs at other critical cancer transitions. In some models, tumors can be detected as early as 1 week after oncogene induction. Tumor induction can be controlled spatially and temporally using cell-type-specific constitutive or inducible gene knockout alone or together with local injection of lentiviruses or other viruses encoding for Cre, oncogenes or tumor suppressor genes. Furthermore, GEMM are immunocompetent and therefore the tumors are subjected to immune selection. Moreover, GEMM are notoriously resistant to immunotherapy, in part because GEMM tumors arise so rapidly, providing limited opportunity for the tumors to acquire mutations that generate T cell antigens. Thus, GEMM provide stringent models to test novel immunotherapy approaches. Comparison of gene expression in early versus late GEMM tumors can be used to identify edited genes in the tumor, corresponding changes in infiltrating immune cells and inferred changes in tumor–immune cell communication.

We recently used single-cell RNA-sequencing of an aggressive Her2+ breast cancer GEMM (Turpin et al, 2016; Zhang et al, 2021b) to identify the genes altered during tumor immunoediting (Zhang et al, 2024). Although GEMM have been used to identify key immunoprotective cells and pathways that are suppressed during immunoediting (DuPage et al, 2012) or immunosuppressive cells and pathways, such as TGFβ production (Donkor et al, 2011), so far our study is the only genome-wide study of tumor immunoediting in a GEMM. This breast cancer GEMM has very short latency (tumors were detected within a week of inducing the oncogene and almost all mice developed tumors within a month (Zhang et al, 2021b)) and green fluorescent protein was co-induced with an inducible constitutively active Her2 oncogene. These properties—short latency, an inducible oncogene for synchronizing tumorigenesis and a fluorescent tag to identify emerging

transformed cancer cells—made this GEMM especially suitable for studying immunoediting. These features are lacking in most GEMM. Studying immunoediting in other breast cancer subtypes or other cancers would be aided by engineering additional GEMM with these properties.

In this study, we compared gene expression by scRNA-seq of early tumors harvested 1 week after oncogene induction with late tumors harvested after one month. The original model of tumor editing assumed that gene editing mostly focused on suppressing antitumour immunity rather than on modifying the expression of tumor-cell-intrinsic genes that enhance tumor cell proliferation and promote other malignant properties (Dunn et al, 2002; Khong and Restifo, 2002). The predominance of editing immune-stimulating genes in early tumors was confirmed by our study. In addition, our study suggested that DNA methylation played an important role in immunoediting in this model (Zhang et al, 2024). Editing in tumor cells concentrated on genes that stimulate innate and adaptive immune responses to the tumor rather than on genes that regulate cell-intrinsic malignant properties. Genes involved in innate immune defenses involving type I and type II IFNs and IFN-stimulated genes (ISGs) dominated the genes that were repressed in emerging breast tumors. More than half of the significantly downregulated genes were ISGs. Whereas the importance of innate immune production of type I IFNs in promoting antitumour immunity is well established (Dunn et al, 2006; Zitvogel et al, 2015), and confirmed in this study (Zhang et al, 2024), tumor immunology studies have not paid as much attention to the role of type II IFNs, which can potentially both activate antitumour immunity (by activating antigen presentation and antigen-presenting cells) and suppress it (for example, by inducing PD-L1 (Blank et al, 2004)). Indeed, 39 of the top 40 over-represented gene ontology terms were immune-related and the most significantly repressed immune-related gene ontology terms were those corresponding to innate immunity. Genes whose products participate in pyroptosis or necroptosis were not significantly differentially expressed in early vs late tumors probably because they were not expressed or were only barely expressed in the early tumors or in normal mammary epithelial cells.

This model provided a snapshot of the development of immune cell exhaustion in emerging tumors. CD8+ T cells in tumors

obtained 1 month after oncogene induction had reduced expression of activation and effector proteins, chemokines and adhesion molecules and increased expression of multiple checkpoint inhibitors, characteristics of cells that were exhausted or becoming exhausted (Zhang et al, 2024). Inferred communication between the tumor and infiltrating lymphocytes, based on co-expression of receptors and their ligands, was strongly dysregulated in late tumors. Pathways that were active in early tumors but became inactive in late tumors included both innate immune pathways (such as IFNγ, LIGHT (also known as TNFSF14) and IL-2 signaling) and adaptive immune pathways (such as CD80 costimulation and MHC class II binding). For example, NK cells in early, but not late, tumors expressed IFNγ, which activates macrophages and suppresses tumor growth, which was sensed by a broad range of tumor cell subtypes and most immune cells. Similarly, tissue-resident memory CD8$^+$ T cells expressed LIGHT, a proinflammatory cytokine that stimulates T cells and activates NF-κB to increase innate immune signaling and/or trigger apoptosis (Skeate et al, 2020), which was sensed only in early tumors by all tumor cell and some immune cell subclusters. This study showed that tumor immunoediting develops rapidly to dampen both innate and adaptive immune signaling, disrupt tumor–immune cell interactions and promote T cell exhaustion.

The repression of edited immune genes in GEMM tumors could be reversed by inhibiting DNA methylation using decitabine (DAC). DAC suppressed tumor growth and restored immune control by increasing the numbers and functionality of tumor-infiltrating CD8$^+$ T cells and NK cells, CD8$^+$ tissue-resident memory T cells and CD103$^+$ DCs with cross-priming capacity, and by reducing the number of MDSCs (Zhang et al, 2024). DAC-induced genes in the GEMM, as well as in 4T1 breast cancer and B16F10 melanoma implanted tumors, included key genes for IFN signaling, inflammatory cytokines, pyroptosis and necroptosis, which overlapped markedly with the genes that were shown to be repressed during cancer immunoediting. Perhaps surprisingly, low-dose DAC derepressed genes that are suppressed during tumor editing without increasing the expression of genes that determine alternate differentiation states. This differential derepression might be because the latter set of

genes are more tightly repressed by more extensive histone modifications and thus changes in their chromatin structure are not as easily induced. DAC-treated B16F10 tumors also had increased numbers of tumor cells undergoing inflammatory cell death. Despite the prominence of DAC-induced upregulation of type I IFN pathway genes, the tumor-suppressive effect of DAC against B16F10 tumors was sustained even in IFNα/β receptor knockout (*Ifnar1*$^{-/-}$) mice. Knocking out *Irf3, Irf7, Gsdme* or *Ripk3* in B16F10 tumor cells each reduced DAC's effectiveness at inducing immune control of the tumor, which indicates that the derepression of multiple innate immune pathways helped to restore immune control (Zhang et al, 2024).

Cancer researchers have been wary of activating inflammation, because inflammation has a dual role in cancer—it both promotes cellular transformation and enhances antitumour immunity. Chronic inflammation has a well-established epidemiological role in promoting tumourigenesis (for example, increased cancer predisposition in smokers or individuals with inflammatory bowel disease) and transcription factors that are activated by inflammation, such as NF-κB, AP1 and STAT3, promote cellular transformation (Grivennikov et al, 2010; Zhao et al, 2021). The cancer-promoting role of inflammation was clearly demonstrated by the CANTOS trial, in which patients at risk for cardiovascular disease who were treated with a blocking antibody to the inflammatory cytokine IL-1β had reduced incidence of lung cancer (Ridker et al, 2017). However, inflammation can also augment antitumour immunity because it helps to distinguish tumor tissue from normal tissue. The benefit of increased immunogenicity stimulated by the acute and potent inflammation that accompanies ICD within already formed tumors may outweigh the dangerous carcinogenic effects of chronic tissue inflammation in normal tissues, by enhancing antitumour immunity, but this needs to be studied in patients.

Activation of the inflammatory cell death pathways of pyroptosis and necroptosis in tumor cells triggers ICD, which recruits immune cells into the TME and activates them (Aaes et al, 2016; Zhang et al, 2020) (Box 2). Tumor immunoediting of inflammatory genes could have a more important role in tumors that arise from barrier tissues, such as the skin or mucosal

epithelia, as these are constantly exposed to microorganisms, which prime expression of genes in these pathways. Future studies of immunoediting in GEMM models of cancers, such as melanoma, CRC or lung cancer, that arise in microorganism-exposed tissues may provide a more complete picture of the innate immune pathways that contribute to immune surveillance of newly formed tumors. Microbial exposure that increases a cell's readiness to induce innate immunity may help to explain the overrepresentation of skin and mucosal epithelial tumors in the solid tumors that respond to ICB.

The study of tumor immunoediting in a breast cancer GEMM (Zhang et al, 2024) is just a beginning. Future studies could be used to improve our understanding of T cell activation by emerging tumors and of how early immunoediting drives T cell and NK cell dysfunction, myeloid cell suppression and tumor evasion. Future studies in other inducible GEMMs engineered with short latency could determine whether the prominence of tumor editing of innate and adaptive immune genes rather than genes that promote proliferation and malignancy is a common feature of different cancers and could expand our knowledge of how different tumors respond to immune surveillance or other challenges that might be specific to the tissue in which they arise. Kinetic information could be captured to get a more detailed picture of the evolution of immunoediting and how it affects the function of other cells in the TME to create an environment that supports tumor immune evasion, persistence and growth. More in depth sequencing or sequencing of selected subsets of cells will be needed to study the interactions of the tumor with rare lymphocyte subtypes, myeloid cells or stromal cells in the TME. Spatial transcriptomics would provide a way to understand better the molecular basis of how immune cells are excluded from some tumors and what distinguishes the tumors that are able to coexist with infiltrating immune cells or tolerate the organization of immune cells into tertiary lymphoid structures. Although early human tumors are rarely captured before immunoediting occurs, it may be possible to uncover the gene editing strategies cancers use to evade immune recognition by studying high risk patients genetically predisposed to cancer, who undergo regular screening biopsies.

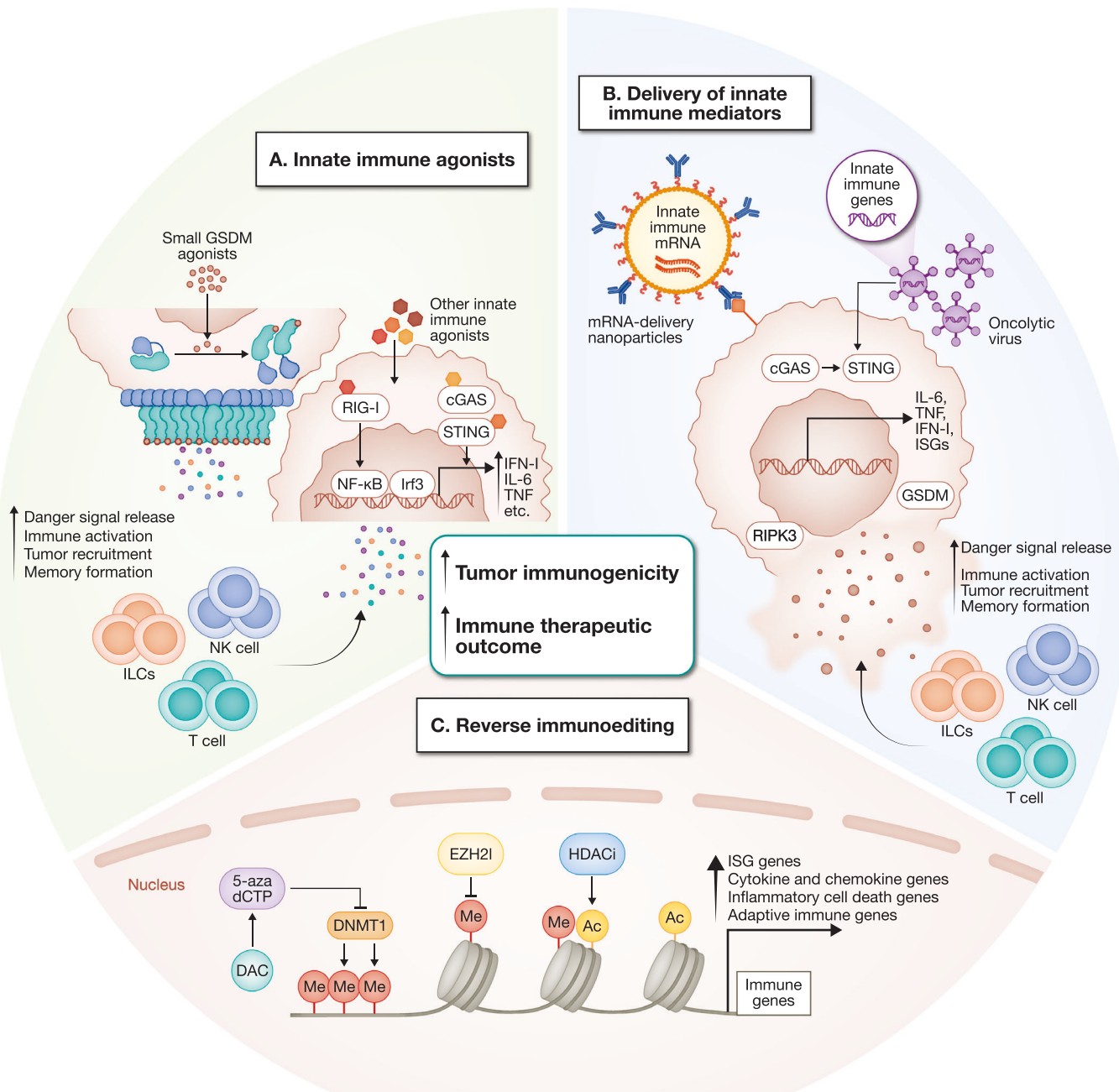

**Figure 4.    Immunotherapeutic strategies to activate innate immunity.**

Tumor immunogenicity can be enhanced by activating innate immune pathways through small-molecule agonists, gene delivery, or epigenetic reprogramming. (**A**) GSDM agonists like GSDMD agonist DMB trigger tumor pyroptosis, releasing danger signals that drive the cancer-immunity cycle, while nucleic acid sensor agonists (RIG-I, cGAS, STING) activate NF-κB and IRF3 to induce type I interferons (IFN-I) and inflammatory cytokines for potent T cell activation. (**B**) Key innate immune mediators such as cGAS/STING, gasdermins (GSDMs), and RIPK3 can be introduced into tumor cells via mRNA nanoparticles or oncolytic viruses to amplify danger signal expression and boost tumor immunogenicity, enhancing cancer immunity. (**C**) Key innate immune genes are frequently silenced by epigenetic modifications during early tumourigenesis. Tumor-targeted epigenetic approaches, including DAC-induced DNA demethylation, EZH2 inhibition (EZH2i)-mediated histone demethylation, or HDAC inhibitors (HDACi), could restore innate immune gene expression and boost tumor immunogenicity.

**Table 1. Cancer therapeutic strategies that activate innate immune pathways**

| Pathways | Signaling outcome | Therapeutic Strategy | Example | | Clinical stage | Ref |
|---|---|---|---|---|---|---|
| cGAS-STING | Secrete type I IFNs, pro-inflammatory cytokines and chemokines to activate antitumor immunity (but also promote tumorigenesis in certain contexts) | STING Agonist | CDN: MK1545 | | Phase II with ICB | (Chang et al, 2022) |
| | | | Non-CDN: DMAXX | | Phase III with Chemotherapy | (Lara et al, 2011) |
| | | Deliver cGAMP | PC7A NP | | Phase I | (Li et al, 2025) |
| | | Block STING degradation | Bafilomycin A | | Preclinical | (Gonugunta et al, 2017) |
| | | Block cGAMP degradation | ENPP1 Inhibitor: STF-1084 | | Preclinical | (Carozza et al, 2020) |
| | | Block dsDNA degradation | TREX1 inhibition | | Preclinical | (Tani et al, 2024) |
| | | STING inhibitor | C-176 | | Preclinical | (Li et al, 2023b) |
| DNA Damage Repair | Increase dsDNA breaks, chromosomal instability, micronuclei and activation of the cGAS-STING pathway and possibly other cytosolic DNA sensors of pyroptosis and necroptosis | Induce DNA damage | Topoisomerase I inhibitors: | Topotecan | Approved by FDA for ovarian and lung cancer, and cervical cancer (with cisplatin) | (Long et al, 2005; ten Bokkel Huinink et al, 1997; von Pawel et al, 1999) |
| | | | | Irinotecan | Approved by FDA for colorectal cancer | (Van Cutsem & Blijham, 1999) |
| | | | Topoisomerase II inhibitors: | Doxorubicin | Approved by FDA for breast, bladder, lymphomas, leukemia and many other cancers | (Blum & Carter, 1974; Sritharan & Sivalingam, 2021) |
| | | | | Etoposide | Approved by FDA for small cell lung cancer (with cisplatin) and testicular cancer (in combination regimens) | (Loehrer, 1991; Noda et al, 2002) |
| | | | Radiotherapy | | Approved by FDA for all major cancer types | (Baskar et al, 2012; Deng et al, 2014) |
| | | Inhibit DNA repair | PARPi inhibitors: (Induce synthetic lethality; their antitumor effects may partially depend on cGAS-STING activation) | Olaparib | Approved by FDA for BRCA-mutated ovarian, breast, pancreatic and prostate cancer | (Golan et al, 2019; Ledermann et al, 2012; Moore et al, 2018; Robson et al, 2017) |
| | | | | Niraparib | Approved by FDA for ovarian cancer | (Gonzalez-Martin et al, 2019) |
| | | | | Rucaparib | Approved by FDA for BRCA-mutated ovarian, prostate cancer | (Coleman et al, 2017; Fizazi et al, 2023) |
| | | | | Talazoparib | Approved by FDA for BRCA-mutated breast cancer | (Litton et al, 2018) |
| | | | DNA-PK inhibitor: M3814 | | Phase II | (van Bussel et al, 2021) |
| | | | ATR inhibitor: Berzosertib | | Phase II | (Yap et al, 2020) |
| | | | ATM inhibitor: AZD1390 | | Phase I | (Durant et al, 2018) |
| | | | WRN Inhibitor: RO7589831 | | Phase II | (Baltgalvis et al, 2024) |
| | | | CHK1/2 inhibitor: Prexasertib | | Phase II | (Giudice et al, 2024) |
| | | | Wee1 inhibitor: Adavosertib | | Phase II | (Maldonado et al, 2024) |
| | | | Rad51 inhibitor: CYT-0851 | | Phase I/II | (Tsang & Munster, 2022) |

**Table 1.** (continued)

| Pathways | Signaling outcome | Therapeutic Strategy | Example | | Clinical stage | Ref |
|---|---|---|---|---|---|---|
| NLRs, other Inflammasomes and Pyroptosis | ICD to release DAMPs, cytokines (particularly IL-1β) and chemokines, to activate antitumor immunity (but also promote tumorigenesis in a context-dependent manner) | Inflammasome inhibitor | Caspase-1 inhibition: | Thalidomide (indirect effect) | Approved by FDA for multiple myeloma, but the contribution of casp1 inhibition to effectiveness in patients is unknown | (Jayabalan et al, 2023) |
| | | | NLRP3 inhibitor: | RRx-001 | Phase III | (Chen et al, 2018) |
| | | | | MCC950 | Preclinical | (Singhal et al, 1999) |
| | | Block IL-1 signaling | IL-1 receptor antagonist: Anakinra | | Preclinical | (Hubner et al, 2020) |
| | | Neutralize IL-1β | IL-1β antibody: Canakinumab | | Phase III | (Ridker et al, 2017) |
| | | Neutralize IL-1α | IL-1α antibody: MABp1 | | Phase III | (Hickish et al, 2017) |
| | | Recombinant cytokine | rhIL-18: SB-485232 | | Phase II | (Tarhini et al, 2009) |
| | | Pyroptosis activator | NLRP3 agonist: BMS-986299 | | Phase I | (Nelson et al, 2025) |
| | | | NLRP1 activator and dipeptidyl peptidase inhibitor: Val-boroPro | | Phase II | (Johnson et al, 2018) |
| | | | GSDME activator: Chemotherapy drugs, targeted therapy drugs and radiation | | FDA-approved therapies. The contribution of GSDME-mediated pyroptosis to effectiveness in patients is unknown. | (Zhang et al, 2021c) |
| | | GSDM-NT delivery | GSDMA3-NT delivery: GSDMA3-NP+Phe-BF3 | | Preclinical | (Wang et al, 2020) |
| | | GSDM Agonist | GSDMD agonist: DMB | | Preclinical | (Fontana et al, 2024) |
| Necroptosis | ICD to release DAMPs, cytokines and chemokines, to activate antitumor immunity (but also promote tumorigenesis in a context-dependent manner) | Necroptosis inducer | Chemotherapy drugs, targeted therapy drugs and radiation | | FDA-approved therapies. The contribution of necroptosis to effectiveness in patients is unknown. | (Meier et al, 2024) |
| | | | Caspase-8 inhibitor/SMAC mimetic: emricasan/birinapant | | Preclinical | (Brumatti et al, 2016) |
| | | | Caspase inhibitor/TLR3 agonist: z VAD/PolyI:C | | Preclinical | (Takemura et al, 2015) |
| | | | Increase z-RNA accumulation: CBL0137 | | Preclinical | (Zhang et al, 2022) |
| | | Necroptosis inhibitor | RIPK1 inhibitor: GSK3145095 | | Phase I | (Harris et al, 2019) |
| Ferroptosis | ICD releasing DAMPs, cytokines and chemokines, which activates antitumor immunity (but also promotes tumorigenesis in a context-dependent manner) | Ferroptosis inducer | Chemotherapy drugs, targeted therapy drugs and radiation | | FDA-approved therapies. The contribution of ferroptosis to effectiveness in patients is unknown. | (Zhou et al, 2024) |
| | | | Sorafenib | | Approved by FDA for HCC, RCC and TC. Its antitumor effect may partially depend on ferroptosis | (Louandre et al, 2013) |
| | | | Erastin Derivatives | | Preclinical | (Yang et al, 2014) |
| | | | GPX4 Inhibitor: RSL3 | | Preclinical | (Yang et al., 2014) |
| | | Ferroptosis inhibitor | Liproxstatin-1 | | Preclinical | (Kim et al, 2022) |

**Table 1.** (continued)

| Pathways | Signaling outcome | Therapeutic Strategy | Example | Clinical stage | Ref |
|---|---|---|---|---|---|
| Immunogenic apoptosis | ICD with calreticulin exposure and DAMP release | Immunogenic apoptosis inducer | Chemotherapy drugs and radiation | FDA-approved therapies. The contribution of immunogenic apoptosis to effectiveness in patients is unknown. | (Sprooten et al, 2023) |
| | | | Oxaliplatin | Phase III, with capecitabine and ICB | (Shen et al, 2025) |
| TLRs | Produce pro-inflammatory cytokines, chemokines and type I IFN | TLR Agonist | TLR7 agonist: Imiquimod | Approved by FDA for basal cell carcinoma (Topical) | (Geisse et al, 2004) |
| | | | TLR9 agonist: ODNs | Phase III, with ICB | (Diab et al, 2025) |
| | | | Attenuated bacteria BCG | Approved by FDA for NMIBC, and its effect partially depends on TLR2/4/9 | (Lamm et al, 1991) |
| RLRs | Produce type I IFN and pro-inflammatory cytokines | RLR Agonist | RIG-I agonist: 5'ppp RNA | Phase I | (Moreno et al, 2022) |
| | | | MDA5 agonist: Poly (I:C) | Phase II | (Butowski et al, 2009) |
| Oncolytic virus | Activate cGAS-STING, RLR, TLR or NLR signaling and trigger ICD | Oncolytic viral infection | Herpesvirus: T-VEC | Approved by FDA for melanoma | (Andtbacka et al, 2015) |
| | | | Adenovirus: H101 | Approved in China for HNSCC | (Xia et al, 2004) |
| | | | Onyx-015 | Phase III | (Bischoff et al, 1996) |
| | | | Vaccinia virus: JX-594 | Phase III | (Abou-Alfa et al, 2024) |
| | | | Reovirus: Pelareorep | Phase III | (Galanis et al, 2012) |

## Activating innate immunity for immunotherapy

In the breast cancer GEMM study and in tumor line implants of diverse tumor types in immunocompetent mice (DuPage et al, 2012; Zhang et al, 2024), the repression of edited immune genes in tumors could be reversed by inhibiting DNA methylation using decitabine (DAC) (Fig. 4C), a drug that is approved to treat myelodysplasia and leukemia in the elderly. The effectiveness of DAC in these solid cancer models was immune mediated and its effectiveness depended on derepressing multiple adaptive and innate immune pathways. Although DAC has failed in multiple small clinical trials of patients with solid tumors, who were heavily pretreated and had not responded to multiple chemotherapy regimens (Linnekamp et al, 2017), its use might be worth reexamining. The key to its success in the mouse models may have been use of a low dose of DAC, which was not cytotoxic and did not alter the cell cycle profile. Cytotoxic doses of DAC that kill the tumor and block its proliferation likely also kill the activated lymphocytes that mediate antitumour immunity and prevent them from clonally expanding. The immunostimulatory potential of DAC in solid tumors in pre-clinical models suggests that its proven clinical effectiveness in hematological malignancies may also involve immune mechanisms. It is also worth studying whether other epigenetic drugs that open chromatin and increase gene expression can safely reverse tumor immunoediting and restore immune surveillance or increase the effectiveness of DAC without causing too much toxicity.

Some traditional cancer treatments, such as radiation and some chemotherapy drugs, can trigger cancer cells to undergo ICD and release danger signals. These agents can also cause severe DNA damage that is unrepaired or CIN, which can lead to release of genomic DNA into the cytosol where it triggers DNA sensors to induce innate immune IFN responses or danger signals to potentiate antitumour immunity. However, these sensors and the molecules that signal these pathways are often not expressed in tumor cells—either because they are not expressed in the corresponding normal tissue of origin (if that tissue is not normally exposed to microorganisms), or because they are epigenetically repressed during immunoediting. Therapeutic strategies that cause tumor cells to express the

final mediators of these key innate immune pathways, either in their precursor or activated states, are in the early stages of development. These strategies include the use of oncolytic viruses or mRNA-loaded lipid nanoparticles (Gujar et al, 2018; Kim et al, 2024; Li et al, 2023a; Raja et al, 2018) (Fig. 4B, Table 1). However, owing to the risk of systemic toxicity, strategies that enable tumor-specific expression of innate immune mediators are desirable. Other therapeutic approaches being developed include small-molecule drug agonists that activate RIG-I, cGAS or STING pathways to stimulate type I IFN production (Fig. 4A, Table 1) (Flood et al, 2019; Jiang et al, 2023; Yum et al, 2020). However, so far, clinical trial results have been disappointing. Incorporating small-molecule agonists into antibody–drug conjugates (ADCs) offers an attractive way to achieve tumor-specific activation by using tumor-targeting antibodies to minimize off-target effects. Most of the ADC drug development for cancer uses chemotherapy drug payloads to kill the tumor. Payloads that induce ICD, such as microtubule inhibitors, or activate Toll-like receptors, which should enhance antitumour immunity, are in clinical development (Han et al, 2024). We are unaware of efforts to design ADCs that release immunogenic oligonucleotides into the cytosol to stimulate intracellular innate immune sensors to trigger IFNs, NF-κB, pyroptosis or necroptosis, but these might induce immune control or enhance ICB responsivity and efficacy. A key advantage of therapeutics that induce danger signals in tumor cells is that they can be effective even when they act in only a minority of tumor cells. Unlike cytotoxic therapies, these approaches rely on localized inflammation to recruit immune cells into the TME and activate and amplify effector and memory immune responses (Wang et al, 2020; Zhang et al, 2020). Unlike the other GSDMs, GSDMD is widely expressed at low levels in most cancer cells. A recent paper identified a small-molecule agonist of GSDMD and showed that it could induce protective antitumour immunity in various cancer models by inducing pyroptosis in a small fraction of cancer cells, without causing apparent systemic toxicity (Fontana et al, 2024) (Fig. 4A, Table 1).

## Concluding remarks and future perspectives

Tumor cell editing allows nascent tumors to evade immune surveillance and survive. Uncovering how specific types of tumors

evade immune surveillance could provide a roadmap to identify novel tumor-specific immunotherapy drug targets to reverse immune evasion and thus broaden the range of solid tumors that respond to immune therapy or cancer vaccines. Whether and how innate and adaptive immunity help keep tumor cells that have colonized distant tissue sites from proliferating and invading the tissue to form macroscopic metastases is an important question about which we know little. Figuring out how colonizing tumor cells are initially restrained and how they succeed in escaping from control could guide future strategies for controlling metastasis.

Danger signals are crucial for distinguishing tumor cells from normal cells, which is required for the full functional activation of antitumour T cells and for establishing long-term memory, both of which are necessary for immune control of tumor growth (Curtsinger and Mescher, 2010; Kroemer et al, 2024). Danger signals can also activate innate and innate-like lymphocytes and phagocytic cells to kill tumor cells that lack tumor antigens or that have repressed tumor antigen expression (Chiossone et al, 2018; Ruf et al, 2023; Van Kaer et al, 2022). The relative effectiveness of different innate immune pathways in inducing antitumour immunity has not been studied in any detail, in part because the molecular underpinnings of the inflammatory programmed cell death pathways that likely sound the strongest alarms are just being uncovered. Although the study of tumor immunology has so far focused on Type I IFN signaling and cytokines as danger signals, Type II IFNs and the programmed cell death pathways that trigger ICD could, potentially, provide even more potent immunostimulation.

There is still a lot that we don't know about how these inflammatory cell death pathways are activated and regulated, especially about the transcriptional regulation of the genes that encode the proteins that participate in these pathways and the post-translational modifications of the inflammatory mediators that regulate their function and how these are modified by tumor editing in different tumor settings. Additionally, the determinants of pathway induction, the intricate crosstalk between different inflammatory cell death modalities, and the mechanisms governing pathway selection remain elusive. Although scRNA-seq and spatial transcriptomics are powerful tools for capturing tumor

evolution and studying tumor immunity, their usefulness is somewhat limited for studying inflammatory cell death since gene expression alone does not signify pathway activation. A more direct approach involves quantifying inflammatory cell death in tumors by tracking the uptake of intravenously injected cell-impermeant dyes, which selectively label dying cells (Fontana et al, 2024; Wang et al, 2020). This simple, yet powerful, tool could provide important information about the activation of inflammatory cell death in the TME and its usefulness in inducing antitumour immunity and guide the evaluation of therapeutic and vaccine approaches that seek to activate these pathways.

What molecules are released from tumor cells dying of pyroptosis or necroptosis are not well defined. The proteins released from dying cells can be cleaved or otherwise modified with functional consequences. In addition to proteins, dying tumor cells likely release small molecules such as ATP, nucleic acids, metabolites and oxidized lipids as danger signals. A comprehensive characterization of these signals will require a combination of "omics" approaches and classical biochemical and cell biology techniques to characterize innate immune danger signals and dissect their functions in tumor immunology. Although the molecules that are released during inflammatory cell death are all thought to amplify inflammation, it remains possible, given the complexity of immune regulation, that some of the released molecules may also exert anti-inflammatory effects (Hayashi et al, 2021). Their combinatorial effects on antitumour immunity versus tumor initiation and malignancy are likely highly context-dependent, necessitating meticulous and systematic dissection in human patients and in cancer models in immunocompetent animals that best mimic human tumors.

Developing therapeutic strategies to induce innate immune responses to cancer is still at an early stage of development. It will be important to figure out practical strategies that induce localized inflammation in the tumor without triggering unacceptable systemic inflammation and toxicity. Expanding our knowledge of these innate immune pathways could also be used to design better adjuvants for more effective tumor vaccines.

## Peer review information

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

## Acknowledgements

The authors thank members of their labs for useful discussions and suggestions. This work was supported by NIH grant R35CA305089 (JL).

## Author contributions

**Zhibin Zhang**: Writing—original draft. **Ying Zhang**: Writing—original draft. **Judy Lieberman**: Conceptualization; Writing—original draft; Writing—review and editing.

## Disclosure and competing interests statement

JL is a co-founder of Ventus Therapeutics.

