## [Peer Review File · The EMBO Journal]

Innate immunity in tumour immunoediting and immunosurveillance

Zhibin Zhang, Ying Zhang, and Judy Lieberman

Corresponding authors: Judy Lieberman (Judy.Lieberman@childrens.harvard.edu), Zhibin Zhang (zzhang16@mdanderson.org), Ying Zhang (ying.zhang@pku.edu.cn)

Review Timeline:

Submission Date:	27th Sep 25
Editorial Decision:	3rd Nov 25
Revision Received:	7th Nov 25
Accepted:	11th Nov 25

Editor: Daniel Klimmeck

Transaction Report:

Please note that the manuscript was previously reviewed at another journal and the reports were taken into account in the decision making process at The EMBO Journal. Since the original reviews are not subject to EMBO Press' transparent review process policy, the reports and author response cannot be published.

Dear Dr Lieberman,

Thank you for submitting the revised Perspective article manuscript, amended according to input by referees during the previous peer-review at another venue. Thank you also for your patience with our feedback. As indicated earlier, we have asked referee #1 to reassess your amended manuscript version for our journal. We have now received comments from this expert, which I enclose below, who states that his-her points are very well addressed.

I am thus pleased to let you know that your Perspective article has been provisionally accepted for publication at the EMBO Journal.

We still need you to consider a number of minor changes and amendments with respect to formatting of the manuscript as indicated below. Further, please introduce the wording changes to the abstract as discussed.

Please let me know any time should you have questions related.

I am looking forward to your feedback on this, and seeing the piece accepted and at production shortly.

Best wishes,

Daniel Klimmeck

Daniel Klimmeck, PhD
Senior Editor
The EMBO Journal

EMBOJ-2025-122583, final formatting adjustments required:

>> Please upload the manuscript text in docx format.

>> add up to five keywords to the manuscript.

>> Authors: please define the corresponding author(s) on the title page.

>> Author Contributions: need to be specified in our online system. Note that CRediT has replaced the traditional author contributions section as of now because it offers a systematic machine-readable author contributions format that allows for more effective research assessment. Please use the free text boxes beneath each contributing author's name to add specific details on the author's contribution.

More information is available in our guide to authors.
<https://www.embopress.org/page/journal/14602075/authorguide>

>> Please add a 'Disclosure and competing interests statement' to the manuscript.

>> Callouts: please add callouts for all figure panels in Fig 2, 3 and 4.

Referee #1:

The authors have done a very good job and have improved the manuscript further addressing all of my concerns.

The authors addressed the remaining editorial issues.

Dear Judy,

Thank you for sending us the updated final version of the perspective article.

I am pleased to inform you that your manuscript has been accepted for publication in the EMBO Journal.

Please note that as this is invited front-half content, OA charges applicable to this article will be covered. Please use the following token - [removed] - when entering the licensing process.

If you have any questions, please do not hesitate to contact me.

Thank you again for your kind contribution to The EMBO Journal, which is much appreciated.

Best regards,

Daniel

Daniel Klimmeck, PhD
Senior Editor
The EMBO Journal